# Semi-Natural and Spontaneous Speech Recognition Using Deep Neural Networks with Hybrid Features Unification

**Ammar Amjad** [1] , **Lal Khan** [1] **and Hsien-Tsung Chang** [1,2,3,4,*]

1 Department of Computer Science and Information Engineering, Chang Gung University, Guishan District, Taoyuan City 33302, Taiwan; ammar.amjad12@gmail.com (A.A.); lal.khan.buzdar@gmail.com (L.K.)
2 Department of Physical Medicine and Rehabilitation, Chang Gung Memorial Hospital, Guishan District, Taoyuan City 33302, Taiwan
3 Artificial Intelligence Research Center, Chang Gung University, Guishan District, Taoyuan City 33302, Taiwan
4 Bachelor Program in Artificial Intelligence, Chang Gung University, Guishan District, Taoyuan City 33302, Taiwan
* Correspondence: smallpig@widelab.org

**Abstract:** Recently, identifying speech emotions in a spontaneous database has been a complex and demanding study area. This research presents an entirely new approach for recognizing semi-natural and spontaneous speech emotions with multiple feature fusion and deep neural networks (DNN). A proposed framework extracts the most discriminative features from hybrid acoustic feature sets. However, these feature sets may contain duplicate and irrelevant information, leading to inadequate emotional identification. Therefore, an support vector machine (SVM) algorithm is utilized to identify the most discriminative audio feature map after obtaining the relevant features learned by the fusion approach. We investigated our approach utilizing the eNTERFACE05 and BAUM-1s benchmark databases and observed a significant identification accuracy of 76% for a speaker-independent experiment with SVM and 59% accuracy with , respectively. Furthermore, experiments on the eNTERFACE05 and BAUM-1s dataset indicate that the suggested framework outperformed current state-of-the-art techniques on the semi-natural and spontaneous datasets.

**Keywords:** spontaneous database; semi-natural database; speech emotion recognition; multiple feature fusion; support vector machine

## 1. Introduction

As a means of expressing emotion, the speech signal is significant in human communication. Therefore, sound has attracted the interest of several organizations working in the domains of human-computer interaction (HCI) [1,2]. Suppose in the framework of HCI, if the machine can identify human emotional states from dialogue, it can adapt adequate actions to interact effectively with a particular individual.

The most frequently utilized approaches of emotional perception fall into two categories: categorical and dimensional [3–5]. The former refers to human emotional classes such as happiness, sadness, disgust, fear, anger, surprise, and neutrality that individuals experience regularly. The latter is also known as a dimensional emotional model since it describes the feeling using a continuous emotional space. The model of emotion that is most often employed is a two-dimensional arousal-valence model [6–8]. These studies apply a categorical emotion descriptions approach to classify people's emotions for SER. In Speech emotion recognition (SER), one of the main elements for identifying the classification performance is to extract the most discriminative features from audio data. However, due to the emotional difference between human emotions and a specific characteristic for emotion classification, the single feature component is taken from one feature fails to increase the efficacy. Therefore, several low-level handcrafted characteristics for SER are frequently employed in enormous studies [9,10]. Furthermore, deep neural networks (DNNs) can

extract more accurate feature representations. Various investigations have retrieved global features from voice data and used them for identification tasks [11–13]. After acquiring a range of audio extracted features, such relevant data must be utilized entirely to increase emotional identification efficiency. For example, past studies [11,14] proved the efficacy of the unification technique in SER. Nevertheless, since these many features extracted are inherently diverse, a fundamental difficulty is figuring out how to properly combine these disparate data for improved classification results. Another difficulty in SER is the integration of numerous characteristics. Numerous earlier studies [15–19] have been published that examined main fusion techniques. Whereas most of the fusion methodologies described earlier achieved appropriate efficiency, they mainly fused numerous features into a single high-level feature set. Then lastly inputted into a classifier that has difficulty learning intrinsic correlations between distinct audio extracted features. An appropriate fusion technique is used with a deep learning framework to learn the most discriminative features and identify high-level connections between numerous audio characteristics for emotional classification. The most significant steps for efficacy enhancement in an SER system are feature extraction, feature unification, and fusion system. So, in the following sections, we will look at a few significant studies that have been done to help us better understand some of the key ideas and strategies in these processes.

### 1.1. Acoustic Features Extraction

Numerous works on the SER system have concentrated on the features extraction process as distinct emotional representations [10–12,20–22]. Acoustic features that have been frequently employed for emotion identification in past years may be classified into two types: low and high-level acoustic features. Prosodic, voice quality [23], spectral, and cepstral features are low-level acoustic features. In [10], voice quality characteristics are obtained within a forty-millisecond frame with a ten-millisecond shifting of the window. In contrast, cepstral-based features are extracted within a twenty-five-millisecond frame with a ten-millisecond window shift. Voice-based Conversational Agent can help adolescents with autism spectrum disorder (ASD) manage their everyday needs and difficulties, including anything from self-care to efficient communication [24]. By codifying PILs and making them queryable in plain language, this research offers a conversational agent to increase health literacy and simplify health information retrieval in Italian [25].

Deep learning approaches have been more popular because of their better ability to train discriminative high-level representations from audio inputs [26–28]. In [29] presented numerous approaches that used neural representations learned from extensive audio datasets to recognize emotions. The presented studies on the Interactive Emotional Motion Capture (MOCAP) dataset [30] outperformed the benchmark recurrent neural network(RNN) and obtained 58% accuracy. Similarly, the researchers presented a hierarchical multimodal framework for the IEMOCAP database with word-level fusion and attention. The presented approach outperforms state-of-the-art techniques with 62% accuracy. A variety of acoustic characteristics were used in the studies listed above to analyze and identify emotional responses. Open-source frameworks such as OpenSmile [31] and deep neural networks like SoundNet [26] are used to extract high- and low-level information, respectively. However, most of the above approaches retrieved low- or high-level characteristics to perform current tasks. It was found that the authors failed to utilize the different types of features for enhanced performance fully. They did not consider the fundamental correlation between the high and low-level features.

### 1.2. Processing of a Variety of Acoustic Features

In order to improve the performance of the classifier, it is essential to include some auditory features. Extensive research has demonstrated that various DNN can obtain a distinct feature of the input speech signal from raw audio datasets for recognition. In [32], the authors introduced a set of approaches for learning features across multiple modalities using a combination of deep learning techniques. In [32], authors introduced several

approaches for combining the various deep learning techniques to extract characteristics across multiple modalities. Numerous research on the CUAVE dataset [33] and the AVLetters dataset [34] showed the most successful audio and visual identification and sharing representation learning. Deep Belief Network (DBN), introduced by Srivastava et al., specified a probability distribution across the multimodal feature space to train the predictive framework of different modalities [35]. Experiments showed that the SVM and Linear Discriminant Analysis (LDA) strategies performed well on information identification and retrieval tasks.

Additionally, specific kernel approaches, such as multiple kernel learning (MKL), have been commonly used in recent years to exploit several properties. The main goal of MKL's is to train a collection of kernels and parameters to get higher identification outcomes [36,37]. Following an extensive investigation into the differences and similarities of MKL algorithms, Nen et al. [36] has categorized and evaluated MKL approaches. Research on actual datasets indicated that employing the MKL method rather than a single kernel was beneficial and might lead to better results. Nilufar et al. devised a unique MKL technique for the Gaussian scale [37]. They used it for large datasets with a high degree of dimension. After rigorous testing on various datasets, it became clear that using MKL in conjunction with a unique technique produced promising outcomes compared to other approaches. However, by employing deep neural networks or MKL, all algorithms discussed above simply used various acoustic characteristics to get better results. They neglected that these characteristics were heterogeneous since they were extracted in response to various parts of the original job, and most were used low-level features. As a result, an effective strategy for overcoming the challenge of unifying diverse feature representation to improve performance is required.

### 1.3. Fusion of Multiple Acoustic Features

A fusion approach for merging features information is required to attain high efficacy. In past research works, various fusion methods have been used to perform speech emotion identification. The fusion approaches may be separated into three parts: (1) feature-level, (2) model-level, and (3) decision-level fusion. First, the feature-level fusion approach concatenates different features into a high-level feature space. After that, the feature map is inputted into a classifier for training to enhance efficiency [38,39]. In [38], developed an asynchronous feature-level-fusion strategy that established a unified hybrid features space model for clustering or identification of multimedia data. The experimental results on two audio and visual emotional datasets confirmed that the suggested strategy achieved much better outcomes. Deep multimodal paradigm introduced to recognize emotional states from the speech in [39]. They used a 3-layer DNN to extract high-level features from audio and text data. They then concatenated two extracted features into one feature vector to fuse them utilizing a DNN approach. The suggested technique obtained good results with the IEMOCAP dataset. However, concatenating numerous feature maps will result in poor accuracy in identification since the noise from each modality is included in the final feature map. Unlike the feature-level fusion technique, the decision-level fusion approach employs a unique method to fuse these characteristics' outcomes. Each feature is autonomous and modeled using a distinct SVM or logistic regression(LR) classifier in the decision-level fusion technique.

In [18], the developed deep hybrid model for audiovisual emotion identification. The DNN approaches were used to obtain characteristics from heterogeneous data in various scenarios. The suggested method's promising performance was achieved using the decision-level fusion technique. With a semi-supervised approach and various neural networks, Kim et al. [15] presented a method for multimodal emotion identification. Multiple deep learning models were used to extract multimodal data from video clips, then combined. Lastly, a decision-level fusion technique known as adaptive fusion produced a competitive identification result. Model-level fusion is another approach that combines various characteristics gathered from several models. A common model-level

fusion strategy concatenates the outputs from hidden units of various neural networks (NN). Model-level fusion strategy develop for the continuous Hidden Markov Model (HMM) classifier in [40]. Large-scale experiments on an extensive collection of ground penetrating radar (GPR) alarms indicated that each feature was employed individually. Furthermore, both features were integrated with equal weights; the model-level fusion attained a promising performance compared to the baseline HMM. Those studies used various fusion procedures to improve performance. However, they failed to notice that the many traits were heterogeneous. Therefore, regardless of whether the fusion technique was utilized, those systems that used heterogeneous characteristics as input data would not provide the best results. Nevertheless, as mentioned above, some of the thoughts and ideas from the studies have influenced our work.

To deal with the issues raised earlier, we investigate the techniques to maximize the utilization of high and low audio features. In addition, we also investigate the methods to maximize the capability of deep learning approaches to fuse numerous sets of information for improved recognition accuracy. In comparison to previous studies on SER, our significant contributions are noted below and explained in the following sections.

- The use of handcrafted and high-level features for SER is not recommended since they are redundant and unconnected. Instead, we introduced a hybrid framework that can successfully generate meaningful feature vectors from the different audio feature sets, eliminating redundant and unrelated information.
- Following an investigation of several fusion strategies, a fusion model based on a DNN is presented to fuse relevant feature descriptions for improved emotion recognition outcomes.
- We evaluate the suggested approach to well-established methods for detecting auditory emotional states. Extensive experimental finding on publically available datasets indicates that our approach gives excellent outcomes, confirming the utility of our technique.

The rest of this article is arranged in the following way. Section 1 reviews some significant and related studies on SER. Section 2 introduces and details the suggested architecture for speech emotion classification. Section 3 summarizes and analyzes the experimental outcomes. Finally, Section 4 contains the findings and recommendations for further study.

## 2. Proposed Model

Figure 1 depicts the suggested framework for stimulated and spontaneous speech emotional classification. Our suggested approach is divided into two steps: (1) Features Extraction and (2) Hybrid Fusing Model. In the following sections, we will go through the processes listed above. The extraction of numerous feature sets from raw audio data in the proposed study used low and high-level acoustic characteristics.

The four different low-level audio characteristics are emphasized in red boxes. In contrast, the two main types of high level audio characteristics are shown in green boxes. One of the main components of the suggested approach is the hybrid unit. The hybrid unit consists of six networks. First, the hybrid unit is designed to address the problem of diversity that exists among the many features extracted from the features unit. Following that, every heterogeneous feature is input into neural networks.

### 2.1. Features Extraction Module

Semi-Natural and spontaneous speech recognition using deep neural networks with feature unification as shown in Figure 1, the first step of the proposed technique is to extract the features from the raw dataset. OpenSmile [31] is a free and open-source toolkit for extracting low-level acoustic characteristics commonly employed in SER. In this study, different low-level acoustic, such as IS10 [41], MFCCs [42],Computational Paralinguistics Challenge (ComParE) [41] and Extended Geneva Minimalistic Acoustic Parameter Set (eGemaps) [43], are extracted from a raw audio databases. The audio feature subset

IS10 contains low acoustic characteristics such as energy, pitch, and jitter, which can be obtained by OpenSmile [31]. The most well-known spectral characteristic is MFCC. It is a widely used technique based on the known variance in the human ear's essential frequency bandwidth. An MFC [44] is composed of MFCCs, coefficients derived from speech data that work together. They were obtained by decorrelating the filter banks energies, composed of triangle filters separated sequentially on the Mel frequency scale [45]. In the same way as OpenSmile extracts IS10, ComParE, MFCCs and eGemaps are obtained by OpenSmile and their accompanying configuration files.

Furthermore, we obtain high-level audio characteristics from DNN due to their improved capability to generate the most relevant feature representations from the audio input. However, because of the limitation of emotional datasets, it is challenging to extract discriminative features from the complicated deep learning networks, which must be well trained. Moreover, many studies commonly report bottleneck characteristics derived from fine-tuned deep learning models for identification problems. In the proposed study, SoundNet [46] and VGGish [47] features are used for emotional classification. SoundNet feature can learn complex natural sound representations from enormous volumes of unlabeled sound data gathered in the environment. Our study uses the SoundNet and VGGish networks as high-level feature extraction that is very effective for speech classification.

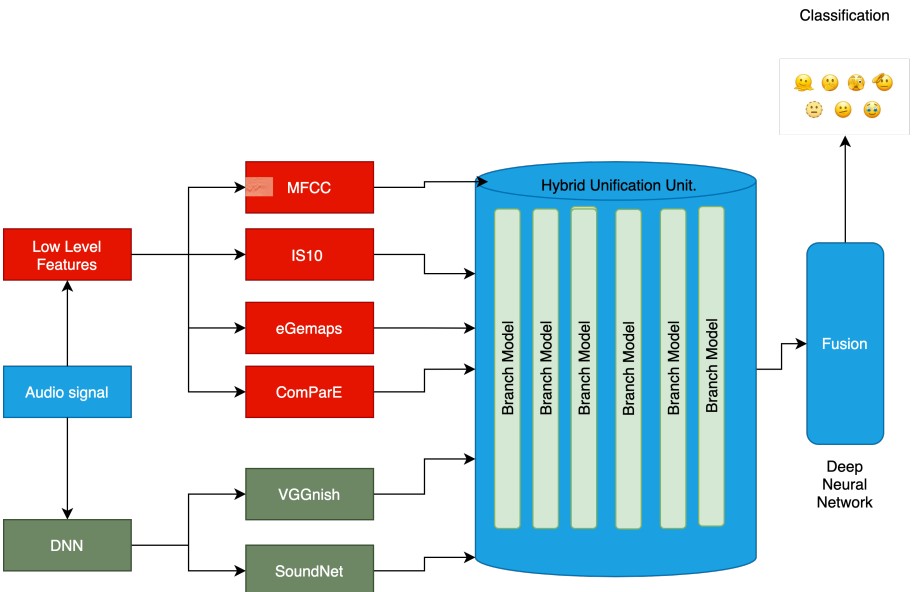

**Figure 1.** The suggested architecture for SER Framework.

As a consequence, by using OpenSmile, we may get a variety of low-level acoustic features. When pre-trained neural networks are used, they provide the SoundNet bottleneck feature and VGGish bottleneck feature, respectively. The features extraction module obtains different acoustic features for emotional classification by using various feature extraction methods. Although, the numerous features are high-dimensional and distinct, with unique patterns in various feature spaces. In order to get better recognition performance, it is not easy to utilize their underlying relationship at the low-level depictions spaces and fuse it. A heterogeneous unification approach is employed to transform different features' heterogeneous space into unified representation space using an unsupervised feature learning approach based on DNN [48,49]. For this reason, the autoencoder structure is constantly being used to learn the new non-linear transformation at a high level from the previously acquired feature representation space. Below is a detailed discussion of the proposed architecture that employs the autoencoder structure and its modifications to produce abstract high-level representations.

### 2.2. Heterogeneous Fusing Model

The features extraction module obtains different auditory characteristics for emotion categorization by using various feature extraction methods. Nonetheless, the numerous characteristics are often high-dimensional and diverse, with unique patterns in various feature spaces. It is difficult to combine their low-level depictions to increase identification accuracy because of the underlying correlation between elements. Combining their low-level depictions to increase identification accuracy is difficult because of the underlying correlation between elements. Therefore, a heterogeneous framework is introduced in previous studies [48,49] to transform the hybrid space of different features into a unified features extraction space using an unsupervised approach based on DNN. Because of its ability to learn the features, the autoencoder architecture is used to learn a new non-linear modification at the high-level spaces from the previously acquired feature extraction spaces. Auto-encoder architecture and its variations used to generate low abstract representations in the presented approach are presented and explored in depth in the subsections.

#### 2.2.1. AutoEncoder

Figure 2 shows an autoencoder, a feed-forward neural network with multiple layers. We define the following with a training sample of p instances $(a_i, b_i) : \{(a_i, b_i) | a_i \in \mathbb{R}^N, b_i \in \{-1, 1\}\}$, S. $i = 1, 2, 3, \ldots\ldots, n$. Where $a_i$ is a feature from the N-dimensional feature space a and $b_i$ is the class to which $a_i$ belongs. The hidden representation $(hl(a_i)) \in \mathbb{R}^M$ in response to an input $(a_i) \in \mathbb{R}^N$ is:

$$hl(a_i) = f(Wt_i a_i + b_i). \tag{1}$$

$Wt_i \in \mathbb{R}^{M*N}$ represents a value of weight, and $b_i$ indicates the bias value in the non-linear transformation function $f$. The rectified linear unit(ReLu) is often employed as the non-linearity after the encoder's final output units. Finally, the system's output decrypts the hidden representation $hl(a_i)$ into a reconstruction $a_i \mathbb{R}^N$.

$$y^i = k(Wt_i hl(a_i + b_i) \tag{2}$$

In Equation (2) $k$ is used as a non-linear activation function.

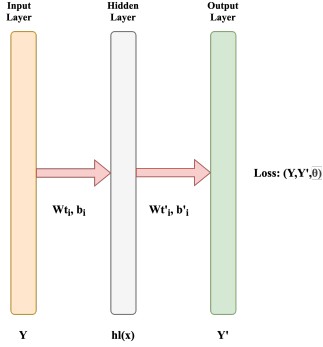

**Figure 2.** An autoencoder's layout.

Thus, the variables $Wt_i, b_i$ denote the relationships between the input and hidden layers, whereas $Wt_i, b_i$ denote the relationships between the hidden and final output layer. Then, a loss function must be specified to reduce the recovery loss $(Y, Y', \theta)$ as alternatively the typical error function:

$$loss(Y, Y', \theta) = ||Y - Y'(\theta)||^3 \tag{3}$$

Learning the auto-encoder entails maximizing the parameter to decrease the reconstructive loss $l(x, x)$ on the training instances. As in the training of NN, stochastic gradient descent is used in small batches.

### 2.2.2. AutoEncoder Denoising

The denoising autoencoder (DAE) [50] is a simple autoencoder extension. Essentially, the goal of DAE is to develop a fundamental auto-encoder that can recreate the original data input after it has been tampered with by adding noisy data. Autoencoders tuned for current recognition tasks may automatically denoise the input and provide more robust feature learning.

### 2.2.3. An Efficient Shared-Hidden-Layer Autoencoder

Identical to the concept of transfer learning, a successful form of the simple autoencoder, called Shared-hidden-layer Autoencoders [49] , was developed solely to share information. Shared-hidden-layer Autoencoders is a technique in which the encoder uses the identical variables for mapping the input and hidden layer as for the mapping but uses separate variables for transformation. Figure 3 shows how Shared-hidden-layer Autoencoders was suggested to reduce the overfitting on both the training and test data. Given the training and test datasets $x_t r$ and $x_{te}$, the two loss functions are as follows:

Training loss $(Y^{tr}, Y', \theta^{tr}) \cong \| Y^{tr} - Y'(\theta^{tr}) \|^2$

Testing loss $(Y^{ts}, Y', \theta^{ts}) \cong \| Y^{ts} - Y'(\theta^{ts}) \|^2$

In the Equation (3) $\theta^{tr} = Wt_i, b_i, Wt^{tr}, b^{tr}$ and $\theta^{t3} = Wt_i, b_i, Wt^{ts}, b^{ts}$ are the train and test span values, respectively. The two functions have identical arguments, $Wt_i$ and $b_i$ that indicate the relationships from the input to the hidden layer, as shown in the above equation. Furthermore, in [49], the corresponding loss function was created to maximize the combined distances for the two sets:

$$loss^{all}(\theta^{all}) = loss^{tr}(Y^{tr}, Y, \theta^{tr}) + lambdaloss^{ts}(Y^{ts}, Y', \theta^{ts}) \tag{4}$$

Lastly, the corresponding objective function is identified as the ultimate objective function:

$$\iota(\theta^{all}) = \min_{\theta^{all}} loss^{all}(\theta^{all}) + \gamma 1(||Wt^{tr}||_1 + ||Wt||_1) + \gamma 2||Wt^{ts}||_2 \tag{5}$$

This study offers an enhanced Shared-hidden-layer Autoencoders approach for generating high-level feature representation from the hidden units. There are two contracts between the standard Shared-hidden-layer Autoencoders paradigm and the enhanced Shared-hidden-layer Autoencoders paradigm.As a result of these changes, the final objective function in Equation (5) and an auxiliary level that exploits advantageous inherent correlations from numerous basic characteristics are enhanced. As shown in Figure 1, the heterogeneous unification module is constructed of numerous branch networks representing various characteristics. Furthermore, in Figure 4, each branch system of the heterogeneous unification module is divided into two sections: the pre-training section and the fine-tuning section. During the processing of pre-training of unsupervised learning, the branch model several hidden units are pre-trained level by level using the different low-level multiple features. The outcome of the previous layer from the encoder functions as the input to the succeeding hidden units, which minimizes the rebuilding error by removing the recreated datasets. For each characteristic, the branch networks that correspond to that feature are distinct from the others. Moreover, the layout of each branch group, including the layout of hidden layers and the size of hidden units, varies from the other branches of the same system. The decoder is replaced by an auxiliary layer shared by all branch networks throughout the supervised fine-tuning process, as seen in Figure 4. The auxiliary layer is used to fine-tune the whole branch network by including supervised data such as classification results or labels. Additionally, the primary objective of the fine-tuning section is to utilize the inherent correlations between those many heterogeneous variables.

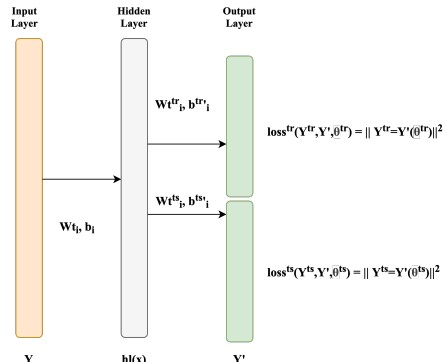

**Figure 3.** The model of an autoencoder.

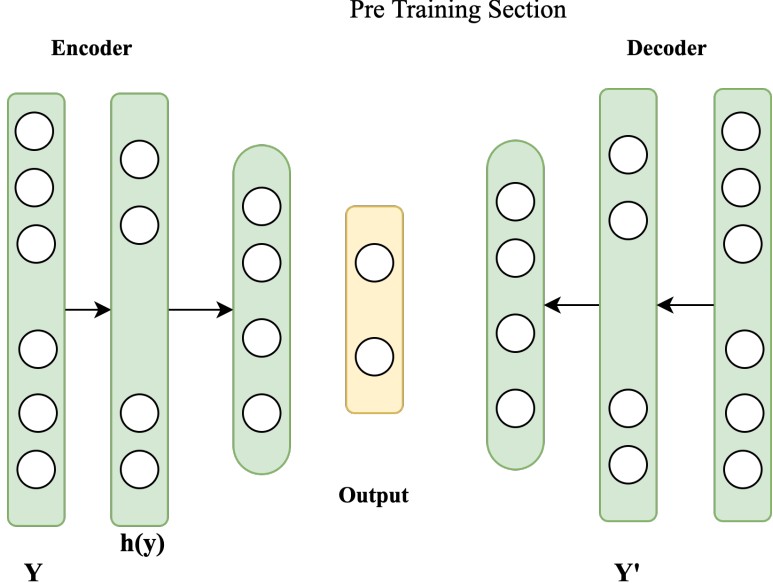

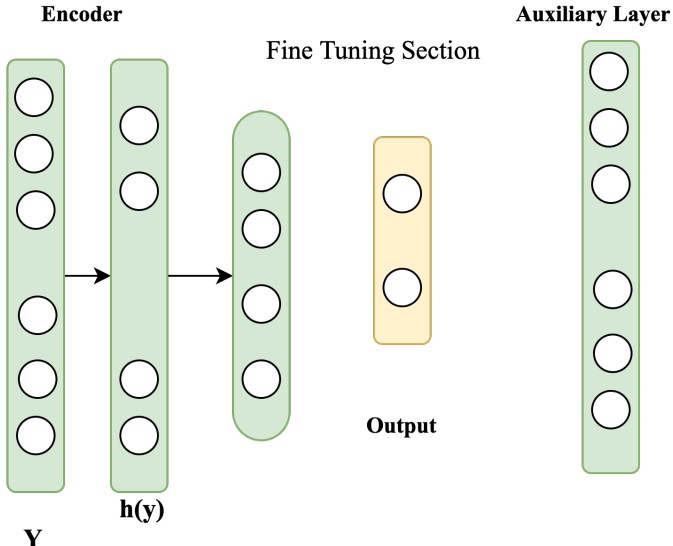

**Figure 4.** The Shared-Hidden-Layer Autoencoder Network framework.

The decoder is replaced by an auxiliary layer shared by all branch networks throughout the supervised fine-tuning process, as seen in Figure 4. The auxiliary layer is used to fine-tune the whole branch network by including supervised data such as classification

results or labels. Additionally, the primary objective of the fine-tuning section is to utilize the inherent correlations between those many heterogeneous variables. Moreover, the suggested system generates the original heterogeneous characteristics into unified forms throughout the fine-tunning, pre-training, and fine-tuning phases. Similarly, stochastic gradient descent is used to learn the hybrid unification unit. Finally, by examining the identification accuracy of the three types of models used in the following experiments, the effectiveness of the suggested architecture is evaluated.

### 2.3. Fusion Network Unit (FNU)

A basic fusion technique is used due to the strength of unifying feature extraction derived from a hybrid unification unit to improve the SER system's efficiency. This research has four levels in the fusion network unit, including one input and three hidden layers. Six modified and unifying high-level characteristics obtained from the branch networks in the hybrid unification unit are fused to generate a combined feature representation, as shown in Figure 1. Next, the FNU identifies the relationships between those unified joint characteristics for the emotional detection challenge, which uses deep neural networks. Finally, the last hidden layer is used to get a feature map used as the overall audio feature extraction in the last hidden layer. Furthermore, an SVM is used as the final classifier to predict our architecture and analyze multiple classifiers' efficacy.

### 3. Experiment

### 3.1. Datasets

### 3.1.1. eNTERFACE05

The eNTERFACE05 [51] audiovisual acting database has 43 participants from 14 different nations. It contains the six emotions of anger, disgust, fear, joy, sadness, and surprise. It includes 1290 video clips. Each audio sample has a sampling rate of 48,000 Hz, a resolution of 16 bits, and a mono channel. Each participant was instructed to listen to six consecutive short tales and designed to elicit a certain feeling. Multiple professionals are hired to determine if the response accurately conveys the desired feeling. Speech utterances are extracted from video recordings of persons speaking in English. The video files are about 3–4 s in length. The dimensions of the original video frames are $720 \times 576 \times 3$.

### 3.1.2. BAUM-1s

The BAUM-1s [16] includes 1222 video clips from 31 Turkish individuals. BAUM-1s database includes the six fundamental classes (joy, anger, sorrow, disgust, fear, and surprise), in addition to boredom and disdain. Additionally, it includes four mental states: uncertain, pondering, focusing, and annoyed. By using a film-induced emotion elicitation, it is possible to attain spontaneous audiovisual emotions. The dimensions of the original video frames are $720 \times 576 \times 3$. As with [16,52], and [53], this study focuses on identifying six fundamental emotions via the use of 521 video snippets. The cropped face pictures in the BAUM-1s dataset.

### 3.2. Experimental Setup

We train our approach using a batch size of 30 and a stochastic gradient descent algorithm with a stochastic mean of 0.9. For fine-tuning, the learning rate is 0.001. Epochs are set to 400 to train the network. The dropout value is set at 0.3 for the fusion technique. The MatConvNet toolkit is used to develop DNNs, and deep models are trained on a single NVIDIA GTX TITAN X GPU with 12GB RAM. We used the LIB-SVM package to run the SVM approach with the linear kernel function and the one-versus-one approach for emotional identification. Following [41], utilized cross-validation methods such as the subject-independent leave one speakers group out (LOSGO) frequently used in the real world. We used the LOSGO approach with five-speaker groups on the eNTERFACE05 and BAUM-1s, both datasets with more than ten individuals. Finally, the average accuracy of test runs is given to compare the performance of all evaluated techniques.

### 3.3. Experimental Results and Analysis

We report experimental findings for local level features and global level features on the semi-natural and natural databases. We ran several experiments using the semi-natural and natural databases to evaluate the suggested emotion classification approach. The accuracy of per-class emotions and overall emotions is compared and analyzed using various approaches in this paper. It is commonly noted that choosing a suitable learning algorithm is vital for an identification framework. Therefore, we initially assess the suggested framework utilizing several classifiers such as K-Nearest Neighbor (KNN), LR, Random Forest (RF), and SVM to determine the most effective classifier. One of the most important aspects of a classification model's performance is selecting hyperparameters. These hyperparameters are calculated in this study based on the accuracy attained on the validation set. In the proposed approach, the overall results of the validation data are used to evaluate the hyperparameters. In addition, we also examined multiple kernel functions to identify the most suitable kernel.

In contrast to the five classifiers shown in Table 1, the SVM algorithm attained the highest outcome for the eNTERFACE05 database. The SVM classifier obtained 76% accuracy for the eNTERFACE05, while the obtained accuracy was 73% for the eNTERFACE05, through MLP. Thus, the classification accuracy of SVM is 21% greater than that of the KNN classifier, which is the most significant difference amongst the five classifiers. In addition, Table 1 shows that MLP and SVM models obtained the same classification accuracy on the seven emotion classes of the eNTERFACE05 database. Therefore, the SVM was considered the best classifier in this research because of its higher classification outcome on the eNTERFACE05 dataset. For example, Table 1 represents that the SVM classifier obtained the highest accuracy for a joy class with 89% accuracy and the lowest for the surprise class with 62% accuracy. Furthermore, compared to the joy class, the classification accuracy for fear and sadness is poor due to the tiny sample sizes of the categories. As seen in Table 1, the same phenomenon occurs when different classifiers are used.

**Table 1.** Comparison of different classifiers' results on the eNTERFACE05 dataset.

|          | SVM  | RF   | LR   | KNN  | MLP  |
|----------|------|------|------|------|------|
| Joy      | 0.89 | 0.85 | 0.81 | 0.74 | 0.87 |
| surprise | 0.62 | 0.25 | 0.45 | 0.59 | 0.75 |
| Anger    | 0.85 | 0.61 | 0.48 | 0.47 | 0.81 |
| disgust  | 0.79 | 0.75 | 0.69 | 0.63 | 0.43 |
| sadness  | 0.73 | 0.66 | 0.52 | 0.55 | 0.87 |
| fear     | 0.71 | 0.69 | 0.81 | 0.34 | 0.68 |
| Total    | 0.76 | 0.63 | 0.62 | 0.55 | 0.73 |

As shown in Table 2, the SVM recognized "surprise" and "joy" with the highest accuracies of 48% and 43% with the BAUM-1s dataset. As shown in Table 1, the eNTERFACE05 dataset contains six emotions, joy, surprise, fear, disgust, sadness, and anger, which are listed with accuracies of 87%, 75%, 68%, 43%, 87%, and 81% respectively with the MLP classifier. The eNTERFACE05 database identified joy with the highest accuracy of 89% with SVM classifier. At the same time anger, and sadness were recognized with the highest accuracies of 85%, and 73% with the SVM classifier, respectively. As shown in Table 2, the MLP recognized sadness and joy with the highest accuracies of 57% and 55% with the BAUM-1s dataset. The BAUM-1s database identified surprise with the highest accuracy of 48%. At the same time, sadness, joy, and disgust were recognized with the highest accuracies of 41%, 43%, and 38% with the SVM classifier, respectively. Compared to the Joy and disgust class, the identification scores for other classes are poor due to the class's sample sizes. As seen in Tables 1 and 2, the same effect occurs when additional classifications approaches are used.

To examine the efficacy of various features, such as ComParE, IS10, MFCCs, SoundNet, and VGGish using the SVM classifier. Table 3 summarizes the classifications outcomes of various characteristics when SVM is used as the classifiers. From the experimental findings in Table 3, we can conclude that various auditory characteristics have varying recognition abilities for the current task. While many other audio features, deep learning features outperform the low-level features that have been frequently utilized and shown to be useful in past studies. For example, as shown in Table 3, the most significant difference is a 32 percent difference in recognizing when comparing outcomes obtained by employing VGGish and ComParE feature sets on eNTERFACE05 dataset. The outcomes demonstrate that deep neural networks have a remarkable capacity to acquire new features through feature learning. According to Table 3, the most outstanding identification accuracy is obtained by utilizing the VGGish and SoundNet features. VGGish and SoundNet features obtained the highest accuracy. SoundNet obtained the 61% average accuracy, while VGGish attained the 67% average accuracy, delicately selected system among multiple CNN frameworks, and better VGGish features' efficiency derived from a well-fine tuned model [47]. Table 4 illustrated that the VGGish and SoundNet feature obtained an average accuracy of 0.36% and 0.35% with the BAUM-1s dataset.

**Table 2.** Comparison of different classifiers' results on the BAUM-1s dataset.

|  | **SVM** | **RF** | **LR** | **KNN** | **MLP** |
|---|---|---|---|---|---|
| Joy | 0.43 | 0.19 | 0.27 | 0.35 | 0.55 |
| surprise | 0.48 | 0.21 | 0.32 | 0.28 | 0.44 |
| Anger | 0.33 | 0.34 | 0.28 | 0.15 | 0.39 |
| disgust | 0.36 | 0.25 | 0.30 | 0.30 | 0.43 |
| sadness | 0.41 | 0.26 | 0.34 | 0.26 | 0.57 |
| fear | 0.32 | 0.29 | 0.38 | 0.13 | 0.48 |
| Total | 0.38 | 0.26 | 0.31 | 0.24 | 0.59 |

Furthermore, to provide a baseline for the proposed strategy, we performed experiments using the BUAM-1s and eNTERFACE05 database. We compared the results to those obtained from earlier studies. We conducted comparison trials between two widely used approaches [18,28,29,54].

**Table 3.** Per-Class Emotions accuracy of multiple features on eNTERFACE05 dataset.

|  | **IS10** | **eGemaps** | **MFCC** | **VGGish** | **SoundNet** | **ComParE** |
|---|---|---|---|---|---|---|
| Joy | 0.38 | 0.39 | 0.45 | 0.71 | 0.64 | 0.39 |
| surprise | 0.37 | 0.55 | 0.39 | 0.73 | 0.59 | 0.32 |
| Anger | 0.39 | 0.58 | 0.32 | 0.58 | 0.79 | 0.29 |
| disgust | 0.25 | 0.47 | 0.40 | 0.79 | 0.68 | 0.52 |
| sadness | 0.35 | 0.49 | 0.35 | 0.65 | 0.53 | 0.29 |
| fear | 0.40 | 0.54 | 0.43 | 0.58 | 0.43 | 0.33 |
| Total | 0.36 | 0.50 | 0.39 | 0.67 | 0.61 | 0.35 |

Additionally, we performed tests on the eNTERFACE05 dataset to compare the suggested technique to previously published methods. We compare two typical current methods [28,29]. According to Table 5, Ref. [29] obtained a performance of 58 percent accuracy on the IEMOCAP dataset. In contrast, in [28] acquired a performance of 62 percent accuracy. In comparison to state-of-the-art approaches, our experimental results show that the proposed technique, in which the improved efficient shared-hidden-layer Autoencoder model serves as the hybrid module's branch model, obtains a higher identification performance than conventional techniques for speech emotion recognition. For example, Ref. [29] presented two models for speech emotion detection that use pre-trained automated speech

recognition (ASR) networks. In addition, scientists explored employing a variety of neural architectures to produce speech characteristics. At the same time, we classify using both low-level and high-level features.

**Table 4.** Per-Class Emotions accuracy of multiple features on BAUM-1s dataset.

|  | IS10 | eGemaps | MFCC | VGGish | SoundNet | ComParE |
|---|---|---|---|---|---|---|
| Joy | 0.41 | 0.39 | 0.32 | 0.75 | 0.52 | 0.45 |
| surprise | 0.14 | 0.21 | 0.09 | 0.15 | 0.26 | 0.14 |
| Anger | 0.12 | 0.27 | 0.07 | 0.25 | 0.19 | 0.31 |
| disgust | 0.39 | 0.34 | 0.39 | 0.44 | 0.66 | 0.45 |
| sadness | 0.32 | 0.26 | 0.27 | 0.48 | 0.33 | 0.05 |
| fear | 0.22 | 0.03 | 0.04 | 0.12 | 0.14 | 0.09 |
| Total | 0.26 | 0.25 | 0.20 | 0.36 | 0.35 | 0.28 |

Furthermore, we performed experiments using the eNterFACE05 and BUAM-1s database to provide a baseline for the proposed strategy. Besides, we compared the results to those obtained from earlier studies. Finally, we conducted comparison trials between two widely used approaches [28,29] On the eNterFACE05 database we can see that [29] obtained efficiency with 58 percent accuracy while [28] obtained efficacy with 62 percent accuracy, as seen in Table 5. Thus, compared to the above-mentioned state-of-the-art approaches, our experimental findings show that the suggested technique obtains superior identification accuracy than previous studies methods for SER. For example, in [29], two models for speech emotion identification were developed to use a pre-trained automated speech recognition (ASR) network. In contrast, we extensively use both low-level and high-level characteristics to classify speech. Table 6 illustrated the performance comparisons of state-of-the-art methods with spontaneous database.

**Table 5.** Comparison Per-class emotions accuracy of state-of-the-art methods with semi-natural database.

|  | Joy | Surprise | Anger | Disgust | Sadness | Fear | Neutral | Total |
|---|---|---|---|---|---|---|---|---|
| [29] | 0.72 | – | 0.59 | – | 0.59 |  | 0.37 | 0.58 |
| [28] | — | – | – | – | – | – | – | 0.62 |
| [8] | 0.83 | – | 0.93 | – | 0.91 | – | 0.89 | 0.89 |
| Our Approach | 0.89 | 0.62 | 0.85 | 0.79 | 0.73 | 0.71 | – | 0.76 |

**Table 6.** Comparison Per-class emotions accuracy of state-of-the-art methods with spontaneous database.

|  | Joy | Surprise | Anger | Disgust | Sadness | Fear | Total |
|---|---|---|---|---|---|---|---|
| [55] | 0.55 | 0.04 | 0.16 | 0.29 | 0.70 | 0.05 | 0.29 |
| [18] | 0.65 | 0.64 | 0.28 | 0.26 | 0.53 | 0.25 | 0.39 |
| [54] | 0.55 | 0.07 | 0.16 | 0.29 | 0.70 | 0.06 | 0.30 |
| proposed approach | 0.75 | 0.15 | 0.25 | 0.44 | 0.48 | 0.12 | 0.36 |

The findings in Table 7 illustrate that proposed framework can learn discriminative relevant data from numerous hybrid features maps and achieve competitive recognition efficiency in SER.

**Table 7.** The per-class emotional performance of elimination experiments with BAUM-1s dataset.

| Approaches | Joy | Surprise | Anger | Disgust | Sadness | Fear | Total |
|------------|------|----------|-------|---------|---------|------|-------|
| proposed-HM | 0.15 | 0.11 | 0.06 | 0.21 | 0.28 | 0.05 | 0.14 |
| Proposed-FN | 0.18 | 0.15 | 0.14 | 0.26 | 0.29 | 0.09 | 18.5 |
| proposed-dae | 0.25 | 0.39 | 0.20 | 0.29 | 0.31 | 0.15 | 0.26 |
| proposed-method | 0.43 | 0.48 | 0.33 | 0.36 | 0.41 | 0.32 | 0.39 |

In Table 7, the technique proposed-HM specifies the removal of the hybrid unification unit from the suggested framework. To be more exact, we combine several extracted characteristics by the feature-based unit into a high-level feature vector that serves as the input to the fusion system unit. The second suggestion, denoted by proposed-FN, indicates that the introduced design lacks the fusion system unit. The third technique, proposed-dae, signifies that the DAE paradigm serves as the various module's branch model. The final technique, proposed- Shared-Hidden-Layer Autoencoder, designates the Shared-Hidden-Layer Autoencoder paradigm as the branch model. The proposed-HM strategy achieves efficiency with a precision of 14%, which is the lowest outcome obtained in the studies. There is a significant difference between proposed-HM outcome and the best 39% obtained using the recommended approach proposed-Shared-Hidden-Layer Autoencoder. It demonstrates the heterogeneous unification module's supremacy across the design. The proposed-FN technique achieves a recognition efficiency of 18.5 percent, demonstrating that the fusion system unit can also improve identification performance by 20.5% compared to the proposed-Shared-Hidden-Layer Autoencoder approach. This testing outcome demonstrates the utility of the fusion system unit developed in this study for producing high-quality identification performance. Additionally, we analyze the various approaches employed in the hybrid module's branch model. As shown in Table 7, the proposed-dae technique achieves a 26 percent efficiency by utilizing DAE as the branch modelin the hybrid unit. However, the findings indicate that improvement is obtained when the Shared-Hidden-Layer Autoencoder framework is used instead of the DAE approach. Two significant findings can be taken from the research mentioned above: (1) the hybrid unification unit may combine different features to increase efficiency, while (2) the fusion system unit effectively improves efficiency in this study.

## 4. Conclusions

In this research, we suggested an SER framework that addressed the issue of diverse acoustic characteristics, which typically degrades the identification efficiency of emotion classification systems. The suggested approach comprises three blocks: (1) features extraction, (2) heterogeneous unification unit, and (3) fusion system unit. The unified and improved features are supplied into the fusion network module instead of various heterogeneous characteristics for the present recognition challenge. The suggested architecture performed well according to experimental findings on semi-natural and natural datasets. Furthermore, it attained competitive identification results compared to many baseline techniques. Besides, the suggested DNN and the methodologies presented in this study may be used in various research fields to combine the best of numerous and heterogeneous features for improved classification performance. We will attempt to run additional tests on other public benchmark datasets in the future to examine our work. The third path of study is to apply this framework to the problem of emotion identification using multimodal information.

**Author Contributions:** A.A., L.K., and H.-T.C. contributed conception, designed the study, and analyzed the results. A.A. wrote the software and executed the experiments. A.A, L.K. and H.-T.C. wrote the first draft of the manuscript. All authors have read and agreed to the published version of the manuscript.

**Funding:** This research received no external funding.

**Data Availability Statement:** The datasets generated for this study are available on request to the corresponding author. http://www.enterface.net/results/ (accessed on 1 December 2021) https://ieeexplore.ieee.org/document/7451244 (accessed on 1 December 2021).

**Conflicts of Interest:** The authors declare that the research was conducted in the absence of any commercial or financial relationships that could be construed as a potential conflict of interest.

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
