# Peer review of "Semi-Natural and Spontaneous Speech Recognition Using Deep Neural Networks with Hybrid Features Unification"

_processes, doi:10.3390/pr9122286_

Round 1
Reviewer 1 Report
This work presents a recognizing semi-natural and spontaneous speech emotions with multiple feature fusion and deep neural networks (DNN). The authors investigate the methods to maximize the capability of deep learning approaches to fuse numerous sets of information for improved recognition accuracy. This paper needs major revisions before the next round review. The detailed review comments are in the following:
- This work tried to apply deep learning methods on some specific areas. However, it is not clear what are the major advantages for what areas?
- English writing should be improved significantly such as "an support", etc.
- In Abstract, what are the Experiments on the eNTERFACE05 and BAUM-1s dataset?
- Both Sections 1 and 2 are showing the previous work results. I would suggest to merge and shorten these two sections. Please highlight your work novelty compared with the current literature. Please see some recent works on different machine learning methods such as Prediction of CO2 absorption by physical solvents using a chemoinformatics-based machine learning model, Data-driven modelling techniques for earth-air heat exchangers to reduce energy consumption in buildings: a review, etc.
- In Section 3.2, why did you choose the Heterogeneous Fusing Model?
- In the experimental section, were the experiments conducted by the authors? If yes, please extend the experimental section.
- All the results are presented in Tables 5 to 7. Any results can be shown in figures? How could you use this model to predict the results? How could the readers find the pros from this model in comparison with other ML models?
- Please delete "6." from "6. Reference".
Author Response
Response to Reviewer 1
(Concern#1)
This work tried to apply deep learning methods on some specific areas. However, it is not clear what are the major advantages for what areas?
Author response: Thanks for the concern.
Author action: Feature extraction module: The main advantage of feature extraction modules is to obtain high-level audio characteristics from DNN due to their improved capability to generate the most relevant feature representations from the audio input. However, because of the limitation of emotional datasets, it is challenging to extract discriminative features from the complicated deep learning networks, which must be well trained. (page#5 line#195)
Heterogeneous Fusing Model : A heterogeneous framework is introduced in previous studies \cite{7244241, 7879177} to transform the hybrid space of different features into a unified features extraction space using an unsupervised approach based on DNN. (page#6 line 222)
Fusion Network Unit (FNU)
A basic fusion technique is used due to the strength of unifying feature extraction derived from a hybrid unification unit to improve the SER system's efficiency. This research has four levels in the fusion network unit, including one input and three hidden layers. Six modified and unifying high-level characteristics obtained from the branch networks in the hybrid unification unit are fused to generate a combined feature representation, as shown in Figure 1. Next, the FNU identifies the relationships between those unified joint characteristics for the emotional detection challenge, which uses deep neural networks. Finally, the last hidden layer is used to get a feature map used as the overall audio feature extraction in the last hidden layer.
Furthermore, an SVM is used as the final classifier to predict our architecture and analyze multiple classifiers' efficacy. (page#8 Line#292)
(Concern#2)
English writing should be improved significantly such as "an support", etc.
Author response: Thanks a lot for the suggestion.
Author action: We updated the manuscript by changing the sentence structure of the manuscript.
Thanks.
(Concern#3)
In Abstract, what are the Experiments on the eNTERFACE05 and BAUM-1s dataset?
Author response: Thanks for the concern.
Author action: We updated the abstract by adding the experiment detailed with results. Following \cite{schuller10b_interspeech}, utilized cross-validation methods such as the subject-independent leave one speakers group out (LOSGO) frequently used in the real world. We used the LOSGO approach with five-speaker groups on the eNTERFACE05 and BAUM-1s, both datasets with more than ten individuals. The average accuracy in the test-runs are finally reported to evaluate the performance of all compared methods.
(Concern#4) Both Sections 1 and 2 are showing the previous work results. I would suggest to merge and shorten these two sections. Please highlight your work novelty compared with the current literature.
Author response: Thank you for pointing this out.
Author action: We updated the manuscript by shortening and merging the two sections(Introduction and proposed work) into one section. Basically, our work is motivated by some opinions and ideas from the research mentioned in the first section. This work proposes a hybrid deep neural architecture to extract distinguished feature representations from heterogeneous acoustic features with two different kind of database(semi-natural and spontaneous). In order to achieve better performance, we employ a multiple-layer deep neural network, which acts as a feature fusion framework to capture the associations between the unified features.
(Concern#5) In Section 3.2, why did you choose the Heterogeneous Fusing Model?
Author response: Thanks for the concern.
Author action: In \cite{7956190}, the developed deep hybrid model for audiovisual emotion identification. The DNN approaches were used to obtain characteristics from heterogeneous data in various scenarios. The suggested method's promising performance was achieved using the decision-level fusion technique. With a semi-supervised approach and various neural networks, Kim et al. \cite {10.1145/3136755.3143005} presented a method for multimodal emotion identification. Multiple deep learning models were used to extract multimodal data from video clips, then combined. Lastly, a decision-level fusion technique known as adaptive fusion produced a competitive identification result. Model-level fusion is another approach that combines various characteristics gathered from several models. A common model-level fusion strategy concatenates the outputs from hidden units of various neural networks (NN). Model-level fusion strategy develop for the continuous Hidden Markov Model (HMM) classifier in \cite{5650350}. Large-scale experiments on an extensive collection of ground penetrating radar (GPR) alarms indicated that each feature was employed individually. Furthermore, both features were integrated with equal weights; the model-level fusion attained a promising performance compared to the baseline HMM.(page#3, line#125)
Those researches adopted different kinds of fusion strategies for better performance, however, they overlooked that the multiple features were heterogeneous essentially. Actually, no matter which of the most suitable fusion strategy was used, those approaches in which heterogeneous features were acted as input data would not achieve the best result.
Our suggested approach is influenced by several ideas gleaned from the above mention studies. First, the suggested approach proposed a hybrid deep neural network(HDNN) for extracting distinct feature representations from hybrid audio features. To improve efficacy, we used a multiple-layer DNN as a framework for extracting the relationships between the unifying characteristics. A detailed description of the suggested architecture is in the next section.
(Concern#6)
In the experimental section, were the experiments conducted by the authors? If yes, please extend the experimental section.
Author response: Thanks a lot for the suggestion.
Author action: We try to revise our manuscript by extended the experimental section. (page#8 Line#292)
(Concern#7) All the results are presented in Tables 5 to 7. Any results can be shown in figures? How could you use this model to predict the results? How could the readers find the pros from this model in comparison with other ML models?
Author response: Thanks for the concern.
Author action :
In this research, we suggested an SER framework that addressed the issue of diverse acoustic characteristics, which typically degrades the identification efficiency of emotion classification systems. The suggested approach comprises three blocks: 1) features extraction, 2) heterogeneous unification unit, and 3) fusion system unit. The unified and improved features are supplied into the fusion network module instead of various heterogeneous characteristics for the present recognition challenge. The suggested architecture performed well according to experimental findings on semi-natural and natural datasets.
In \cite{lakomkin2018reusing, 8967041,7956190,DBLP:journals/corr/abs-1803-11508,DBLP:journals/corr/abs-1805-08660}, used deep learning models with high level features and experiment was performed on acted database. In contrast, we extensively use low-level and high-level characteristics to classify speech with both spontaneous and semi-natural databases. The findings in Table 7 illustrate that the proposed framework can learn discriminative relevant data from numerous hybrid features maps and achieve competitive recognition efficiency in SER.
(Concern#8) Please delete "6." from "6. Reference".
Author response: Thank you for pointing this out.
Author action : We updated the manuscript by deleting 6 from subsection reference. (Page#13 Line#461).
Reviewer 2 Report
The paper proposes a Deep Neural Networks approach for speech emotion recognition able to address the issue of diverse acoustic characteristics, which typically degrades the identification efficiency of emotion classification systems.
The paper is attractive and well organized, but it could be improved by adding background notes on possible applications to use the proposed approach. As an example, conversational systems and social robots may highly benefit of speech emotion recognition, thus authors could cite the following recent works and suggest how the proposed approach can improve conversational system:
- Cha, Inha, et al. "Exploring the Use of a Voice-based Conversational Agent to Empower Adolescents with Autism Spectrum Disorder." Proceedings of the 2021 CHI Conference on Human Factors in Computing Systems.
- Minutolo, Aniello, et al. "A conversational agent for querying Italian Patient Information Leaflets and improving health literacy." Computers in biology and medicine(2021): 105004.
- Caggianese, Giuseppe, et al. "Discovering Leonardo with artificial intelligence and holograms: A user study." Pattern Recognition Letters131 (2020): 361-367.
Moreover, authors should add some implementation details on the configuration of DNN networks, such as batch size, number of epochs, learning rate, number of parameters, and so on.
Furthermore, as also specified by the authors, the proposed approach could be applied in various fields. Some quantitative information about the hardware resources used and the response times obtained could help the scientific community to reapply the described techniques and evaluate their use in particular application fields. Do you need a workstation? Do you need to be connected to a remote processing server? Would it be possible to apply these techniques on edge?
Author Response
Response to Reviewer#2
(Concern#1) The paper is attractive and well organized, but it could be improved by adding background notes on possible applications to use the proposed approach. As an example, conversational systems and social robots may highly benefit of speech emotion recognition thus authors could cite the following recent works and suggest how the proposed approach can improve conversational system.
Cha, Inha, et al. "Exploring the Use of a Voice-based Conversational Agent to Empower Adolescents with Autism Spectrum Disorder." Proceedings of the 2021 CHI Conference on Human Factors in Computing Systems.
Minutolo, Aniello, et al. "A conversational agent for querying Italian Patient Information Leaflets and improving health literacy." Computers in biology and medicine (2021): 105004.
Caggianese, Giuseppe, et al. "Discovering Leonardo with artificial intelligence and holograms: A user study." Pattern Recognition Letters131 (2020): 361-367.
Author response: Thanks a lot for the suggestion.
Author action : We updated the manuscript by citing the following papers into introduction section. (Page#13 Line#461).
(Concern#2) Moreover, authors should add some implementation details on the configuration of DNN networks, such as batch size, number of epochs, learning rate, number of parameters, and so on.
Author response: Thanks a lot for the suggestion.
Author action : We train our approach using a batch size of 30 and a stochastic gradient descent algorithm with a stochastic mean of 0.9. For fine-tuning, the learning rate is 0.001. Epochs are set to 400 to train the network. The dropout value is set at 0.3 for the fusion technique.(page#9 line #331)
(Concern#3) Furthermore, as also specified by the authors, the proposed approach could be applied in various fields. Some quantitative information about the hardware resources used and the response times obtained could help the scientific community to reapply the described techniques and evaluate their use in particular application fields. Do you need a workstation? Do you need to be connected to a remote processing server? Would it be possible to apply these techniques on edge?
Author response: Thanks for the concern.
Author action : The MatConvNet toolkit is used to develop DNNs, and deep models are trained on a single NVIDIA GTX TITAN X GPU with 12GB RAM. We used the LIB-SVM package to run the SVM approach with the linear kernel function and the one-versus-one approach for emotional identification.
Round 2
Reviewer 1 Report
Can be accepted.